# Interaction of a Solitary Wave with Vertical Fully/Partially Submerged Circular Cylinders with/without a Hollow Zone

**Chih-Hua Chang** [1,2]

1   Department of Information Management, Ling-Tung University, Taichung 408, Taiwan;
    changbox@teamail.ltu.edu.tw
2   Natural Science Division in General Education Center, Ling-Tung University, Taichung 408, Taiwan

**Abstract:** In this article, a three-dimensional, fully nonlinear potential wave model is applied based on a curvilinear grid system. This model calculates the wave action on a fully/partially submerged vertical cylinder with or without a hollow zone. As basic verification, a solitary wave hitting a single fully or partially submerged circular cylinder is tested, and our numerical results agree with the experimental results obtained by others. The influence of cylinder immersion depth and size on the wave elevation change on the cylinder surface is considered. The model is also applied to investigate the wave energy of a solitary wave passing through a hollow circular cylinder to determine the effect of the size and draft on the wave oscillating in the hollow zone.

**Keywords:** solitary wave; fully nonlinear wave; three-dimensional wave; partially submerged cylinder; hollow circular cylinder

---

## 1. Introduction

Upright pillars are often used as the foundation of marine platforms. An abundance of studies have been conducted on the waves diffracted by vertical circular cylinders. An early focus was the analysis of linear waves diffracted by vertical cylinders in water. To address this topic, many researchers considered waves in water of infinite depth, including the early studies by Havelock [1], Ursell [2], and MacCamy and Fuchs [3], and the more recent work of Finnegan et al. [4]. Other researchers have focused on considering situations in water of finite depth for bottom-mounted, fixed-floating, or free-floating cylinders, including Miles and Gilbert [5], Garrett [6], Black et al. [7], Molin [8], Yeung [9], Sabuncu and Calisal [10], Williams and Demirbilek [11], Bhatta and Rahman [12], Bhatta and Rahman [13], Bhatta [14], Jiang et al. [15], Li and Liu [16], and Ghadimi et al. [17]. In addition, a twin-cylinder modeled as a wave energy converter device was analyzed by Xu et al. [18]. These studies have focused on the analysis of linear waves with circular cylinders. When considering nonlinear waves, it is relatively difficult to find analytical solutions. Taylor and Hung [19,20] investigated the second-order diffraction forces on a vertical cylinder for regular waves or bichromatic waves. A closed-form solution was derived by Kriebel [21] for the velocity potential resulting from the interaction of second-order Stokes waves with a large vertical circular cylinder. The phenomenon of nonlinear waves with a circular cylinder has been analyzed primarily by numerical (e.g., Kim [22]) or experimental (e.g., Yan [23]) methods.

When waves approach the shore, nonlinear and shallow-water wave characteristics will predominate. To protect coastal facilities, engineers must understand the influence of shallow-water waves. One frequently used nonlinear water wave model is the solitary-wave model. To simplify this physical problem, it is simulated as a long wave with a single hump. In this article, we apply a

numerical model to calculate the interaction of a solitary wave with an immersed vertical cylinder that is or is not hollow.

Many studies have been conducted using the depth-averaged method to determine the interaction of a three-dimensional (3D) solitary wave with vertical cylinders. A shallow-water wave concept and a depth-averaged model have been used to simplify this 3D problem into two dimensions. For example, Wang et al. [24] combined a generalized Boussinesq (gB) model with horizontal curvilinear coordinate grids to study the scattering of a solitary wave as it meets a circular cylinder. Later, Yates and Wang [25] used gauges located around a cylinder to conduct an experiment to measure solitary wave elevations and observe their evolution. Some researchers have investigated structures using cylinder arrays. For example, in their study, Wang and Jiang [26] used a gB model for dual cylinders. Zhao et al. [27] built a mesh system with Cartesian unstructured grids and an O-shaped body-fitted grid, and solved the gB equation using the finite element method to explore the interaction between solitary waves and either two or four cylinders that make contact with the seabed. Neil et al. [28] established curvilinear coordinate grids to solve gB and Green–Naghdi (GN) equations for situations involving a solitary wave passing through three cylinders arranged one behind the other. However, all the above studies were restricted by their application of Boussineq-like models. Experimental work on the interaction of solitary waves with multiple cylinders is rare. One example is the work by Yuan and Huang [29], who measured the solitary wave forces on an array of vertical cylinders.

To theoretically analyze the problem of a solitary wave versus a vertical circular cylinder, Isaacson [30] used diffraction theory. In recent years, with the advances in computer technology, 3D, fully nonlinear, inviscid, or viscous wave models have gradually been developed. Many researchers have numerically investigated the solitary waves radiated by circular cylinders. Isaacson [31] and Isaacson and Cheung [32] applied the boundary integral method to investigate the interaction of a solitary wave with a vertical circular cylinder. Ohyama [33] used the boundary element method (BEM) to solve potential 3D wave equations for a solitary wave acting on a huge vertical cylinder. Using a similar method with curvilinear grids, Yang and Ertekin [34] analyzed fully submerged cylinders hit by nonlinear waves, including solitary and Stokes waves. Later, other researchers began to consider the fluid viscous effect and used Navier–Stokes equations to solve this problem. For example, Wang and Huang [35] employed the SIMPLER algorithm [36] to solve Navier–Stokes equations and used the Marker and Cell method to trace free-surface particles in a solitary wave diffracted by a vertical square cylinder. Mo [37] solved Navier–Stokes equations and combined the Cartesian grid with a quadrangular irregular mesh surrounding a single cylinder or three cylinders. Cao and Wan [38] performed a numerical simulation based on the OpenFOAM code to solve Reynolds-averaged Navier—Stokes (RANS) equations for the run-up of a solitary wave on a circular cylinder with different incident wave heights and circular radii. Almost simultaneously, Leschka and Oumeraci [39] and Leschka et al. [40] applied the OpenFOAM code to simulate a solitary wave traveling through three or more cylinders in a staggered arrangement. A particle-in-cell Navier–Stokes solver was developed by Chen et al. [41] to simulate a solitary wave travelling through a single cylinder or a group of eleven cylinders. However, most of these papers discuss the diffraction of solitary waves by bottom-mounted cylinders.

In recent years, the development of offshore marine resources has required platforms for floating bodies, so attention has been paid to the study of waves affected by fixed or free-floating structures. Kang et al. [42] extended the $\sigma$-coordinate method (developed by Lin, [43]) to solve a 3D Navier–Stokes model using the immersed boundary method to deal with solid boundaries and to solve the problem of nonlinear waves interacting with a fully or partially submerged circular cylinder. A BEM approach was applied by Zhou et al. [44] to solve 3D Euler equations for fixed/floating and fully/partially submerged circular cylinders. In their report of the model results, the authors did not discuss the effect of the immersion draft. Chen and Wang [45] developed a 3D fully nonlinear wave model using multi-block grids to solve the problem of a solitary wave interacting with a half-submerged circular cylinder. These authors also measured the wave elevations near the front and rear of the cylinder for comparison

with their numerical solutions. There has been scant discussion of the case of a cylinder with a gap between its bottom and the seabed.

This article also presents an analysis of the water-level evolutions of a solitary wave passing through a hollow circular cylinder. A wave passing through a hollow structure causes the water level to oscillate in the hollow region, which is relevant to the research of wave energy development. The pattern of waves arriving near shore is often similar to that of shallow water wavers (such as solitary waves or Cnoidal waves). Although a solitary wave is nonperiodic, the scenario can still be simulated as multiple yet individual solitary waves (solitons) that are continuously interacting with coastal structures. Wave energy can be converted into electrical energy in many ways, with the oscillating water column (OWC) being one of the most common. For example, a dielectric elastomer generator (DEG) device developed by Scottish and Italian scientists (https://newatlas.com/dielectric-elastomer-generator-wave-power/58465/) basically consists of an anchored vertical circular cylinder in which a column of air is trapped. The head of the cylinder is sealed with a rubber membrane, and the base is open to the surrounding ocean. The efficient conversion of the power of the water column to drive a propeller in the air chamber to obtain electrical energy is a structural design problem. Basically, if vertical two-dimensional (2D) consideration is taken, the OWC mechanism can be simplified as a problem of the interaction between a wave and two vertical piercing plates. However, this article focuses on the 3D problem and does not review the 2D literature.

In 1970, Garrett [46] was likely the first to explore analytical solutions for small-amplitude waves incident on a hollow cylinder partially immersed in water of finite depth. Zhu and Mitchell [47] revisited Garrett's work using a different approach, which had the advantage of requiring less work in the analysis. The authors also noted that there are two key considerations to determining the maximum wave energy of the OWC: The properties of the OWC chamber and the location of the OWC in the ocean. Later, Simon [48] and Miles [49] used a variational technique to analyze the radiation of surface waves from a submerged cylindrical duct. Harun [50] solved the mild-slope equation for the OWC problem to obtain an analytical solution for a linear long wave diffracted around a hollow cylinder. Recently, a twin-cylinder wave energy device was considered by Xu et al. [18]. Mavrakos's research group conducted a number of experimental and numerical studies of the hydrodynamic problem of the OWC device where regular waves act on an arrangement of a single concentric cylinder or a set of compound concentric cylinders [51–53]. Another possible OWC device that consists of two concentric cylindrical shells placed in water has been studied by Shipway and Evans [54] and McIver and Newman [55]. A floating hollow cylinder placed above a solid bottom cylinder was investigated by Hassan and Bora [56,57]. The type, shape, size, draft depth, and position of the chamber (such as its distance from shore) are known to affect the elevation of the water column in the hollow cylinder. Previous experience indicates that even a simple hollow cylinder with different design factors can influence the wave diffraction around the cylinder and the oscillation in the column. Understanding the wave oscillation behavior in a column is the most important issue for controlling the OWC. In recent years, many 3D OWC studies have been conducted. For example, Simonetti et al. [58] used OpenForm to calculate 3D Navier–Stokes equations by large-eddy simulation to investigate the problem of wave and OWC interaction. Kamath et al. [59] used REEF3D software (an open-source computational fluid dynamics program) to calculate a RANS model to explore the interaction between periodic waves and an OWC and to compare the differences between the 2D and 3D results. Lee et al. [60] also used REEF3D to calculate the interaction of regular waves with a 3D OWC device. These authors noted that the efficiency of the current OWC system requires further study by scientists in many fields. Kim et al. [61] conducted experimental work on regular waves passing through two concentric cylinders. They measured the wave heaves within the ring region defined by the cylinders and analyzed the interaction between a solitary wave and an OWC device. In addition to understanding the oscillation effect, with respect to the design of an OWC device for disaster prevention, it is important to simulate the impact of an extreme wave (solitary wave) on OWC facilities to understand the status of the impacting wave.

To this end, this article considers the influence of the height of the solitary wave, the size of the hollow cylinder, and the draft on the wave diffraction and wave oscillation in hollow regions.

Overall, when a wave propagates to the continental shelf, the shallow-water characteristics of the wave are significant. It is surprising that only a few studies have investigated the interaction of a solitary wave with cylinders with a draft effect. Thus far, it seems that the analysis of the interaction between a solitary wave and OWC is rare. Consequently, in this paper, a 3D fully nonlinear wave model is applied to analyze the cases of a solitary wave hitting a solid or hollow circular cylinder with different drafts. The results obtained in this study fill this knowledge gap. From an engineering point of view, this article also has two main applications: (1) analysis of a long wave encountering the circular support cylinder (pile) of a marine platform and (2) simulation of the heave and drop of water levels when a long wave passes through an OWC device (a hollow cylinder).

## 2. Flow-Field Equations

A 3D free-surface flow region with a vertical cylinder is considered, as illustrated in Figure 1. The fluid is assumed to be inviscid, incompressible, and the motion is irrotational. The model formulations are based on a set of fully nonlinear potential flow equations; all variables are made dimensionless by introducing the referencing length scale $H^*$, the undisturbed water depth, the velocity scale, $\sqrt{gH^*}$ (here $g$ is a constant due to gravity), and $\sqrt{H^*/g}$ as the time scale. In this model, the right-hand referencing frame has its $x$-axis pointing in the positive (right) direction, the $y$-axis is expanding laterally, and the $z$-axis is pointing up; the coordinate origin is located at the water level of the undisturbed fluid region. The non-dimensionalized initial-boundary-value problem can be formulated as the governing equation, initial condition, and associated boundary conditions. These equations can be referenced in many books on classical wave mechanics, such as Stokers [62]. These equations are listed as follows:

$$\phi_{xx} + \phi_{yy} + \phi_{zz} = 0, \text{ in the flow-field region} \tag{1}$$

$$\phi_z = \zeta_t + \phi_x \zeta_x + \phi_y \zeta_y, \text{ at } z = \zeta \tag{2}$$

$$\phi_t + (\phi_x{}^2 + \phi_y{}^2 + \phi_z{}^2)/2 + \zeta = 0, \text{ at } z = \zeta \tag{3}$$

$$\Omega_t \pm \sqrt{(1+\zeta)}\Omega_x = 0, \text{ at x-direction lateral boundaries} \tag{4}$$

$$\frac{\partial \Omega}{\partial \widetilde{n}} = 0, \text{ on the bottom, sidewalls, and cylinder surface} \tag{5}$$

where $\phi$ is the velocity potential function. Subscript characters with the coordinates denote the partial differentiation; $z = \zeta$ is the wave surface, $\Omega$ stands for either the $\phi$ or $\zeta$, the $\pm$ sign in Equation (4) indicates the lateral boundary used for right (+) and left (−) outgoing conditions, and $\widetilde{n}$ in Equation (5) is the solid surface unit normal vector of fluid. According to Schember [63] and Wang [24,64], the right-moving solitary waveform ($\zeta$) and the depth-averaged potential function $\overline{\phi}$ are expressed as:

$$\zeta = \left\{\text{sech}^2 k(x - Ct - X_0) + A_0 \text{ sech}^4 k(x - Ct - X_0)\right\}/(1 + A_0) \tag{6}$$

$$\overline{\phi}(x) = \sqrt{4A_0/3}\tanh k(x - Ct - X_0) \tag{7}$$

where $k = \sqrt{3A_0/[4(1+A_0)]}$, wave celerity is $C = \sqrt{1+A_0}$, $A_0$ = the incident wave height, and $X_0$ = the wave's starting position. Wu [65] derived the relation between the potential function and its average value in the flow region as:

$$\phi(x, z) = \overline{\phi} - A_0(\frac{1}{3} + z + \frac{z^2}{2})\overline{\phi}_{xx} + O(\varepsilon^5) \tag{8}$$

in which, $\varepsilon$ = water depth/wavelength. Substituting Equation (7) into Equation (8) for $t$ = 0, we get:

$$\phi(x,z) \;=\; \sqrt{4A_0/3}\tanh k(x-X_0)\left\{1 + A_0 k^2(\frac{2}{3} + 2z + z^2)\text{sech}^2 k(x-X_0)\right\} \tag{9}$$

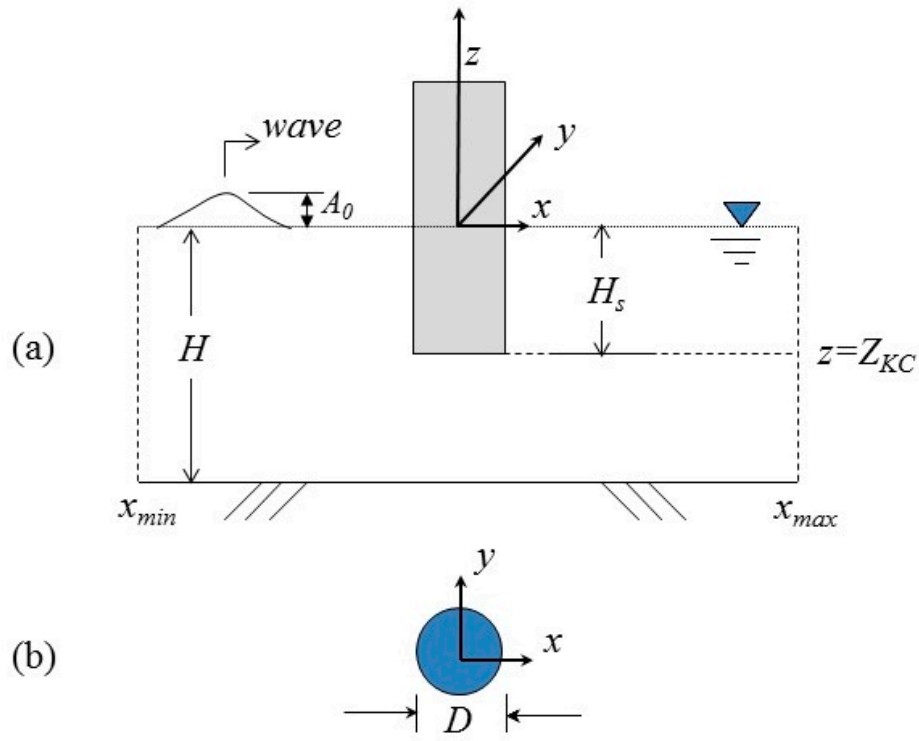

**Figure 1.** Schematic diagram of a wave passing through a vertical circular cylinder. (**a**) side view and (**b**) top view.

Therefore, Equation (6) for $t$ = 0 and Equation (9) are specified as the initial wave profile and potential function in one *x-z* plane, respectively, which are equal in all the *x-z* planes and form the initial 3D conditions.

When calculating waves, there are many methods available for obtaining a 3D solution to the potential energy flow equation, such as the high-order spectral (HOS) method of Dommermuth and Yue [66], which has attracted much attention in recent years. Its advantage is that it can be used for the calculations of a large range of water types. As such, researchers have used the HOS method to calculate a large-scale wave field in combination with computer fluid dynamics to calculate details regarding areas of a wave that encounter a structure (Zhuang and Wan [67]). This article does not focus on innovations in calculation methods, and does not compare our proposed method with others. The numerical model of our work is based on the developmental work of Chang and Wang [68], wherein a transient boundary-fitted curvilinear grid system is used to conform to the moving free surface. The boundary-fitted curvilinear grid was first proposed by Thompson [69]. Here, the explicit–implicit hybrid finite difference scheme is adopted to solve the complete nonlinear free- surface boundary conditions, and discretely solve the internal flow field using the 15-point central difference method in each local computational element. For a detailed explanation of the grid generation process and the finite difference method applied in this numerical method, please refer to the Appendix A.

## 3. Validations

First, the structure of a single-bottomed mounted circular cylinder is considered, and the computation results are compared with those obtained in the experiment conducted by Mo [37]. In Mo's experimental setup, the water depth was 0.75 m, the solitary wave height was 0.3 m, and the cylindrical diameter was 1.22 m. In terms of the dimensionless scale of the water depth, the reference length is a water depth $H = 1$, an incident wave height $A_0 = 0.4$, and a cylindrical diameter $D = 1.63$. As the cylinder is bottom-mounted, $H = H_s = 1.0$. This case is typically adopted to test a 3D nonlinear water wave model. Mo [37] examined this case both experimentally and numerically, and provided abundant data for a solitary wave encountering a single cylinder or multiple piles. First, to verify our model, it is compared with that used by Mo. The calculation range is $(x_{max}, x_{min}) = (-30, 30)$ and $(y_{max}, y_{min}) = (-30, 30)$. This study used six wave-gauge positions (see Figure 2) around the cylinder (as shown in Figure 3). In Figure 3, $t'$ is the time shift with respect to the peak time at Gauge 4. Although our model does not consider fluid viscosity, generally, it can efficiently identify the consistent trends at all six gauge positions.

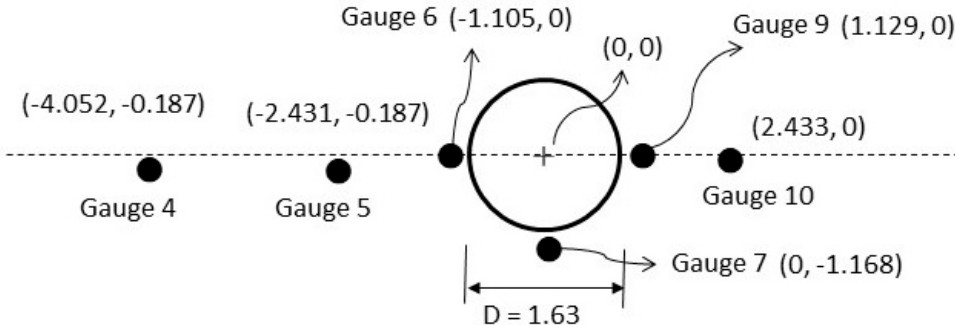

**Figure 2.** Plane view of wave-gauge locations for the experiment by Mo et al. (2010) in a dimensionless scale.

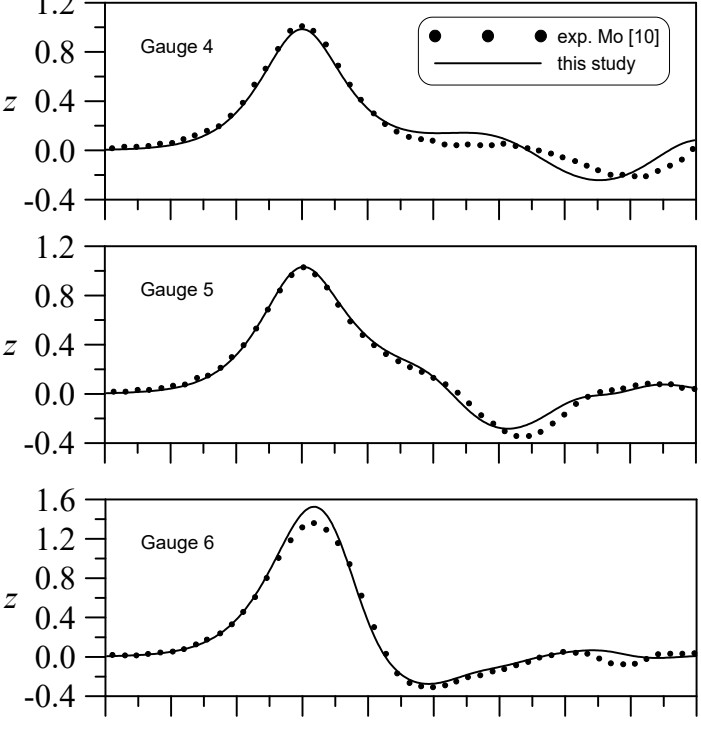

**Figure 3.** *Cont.*

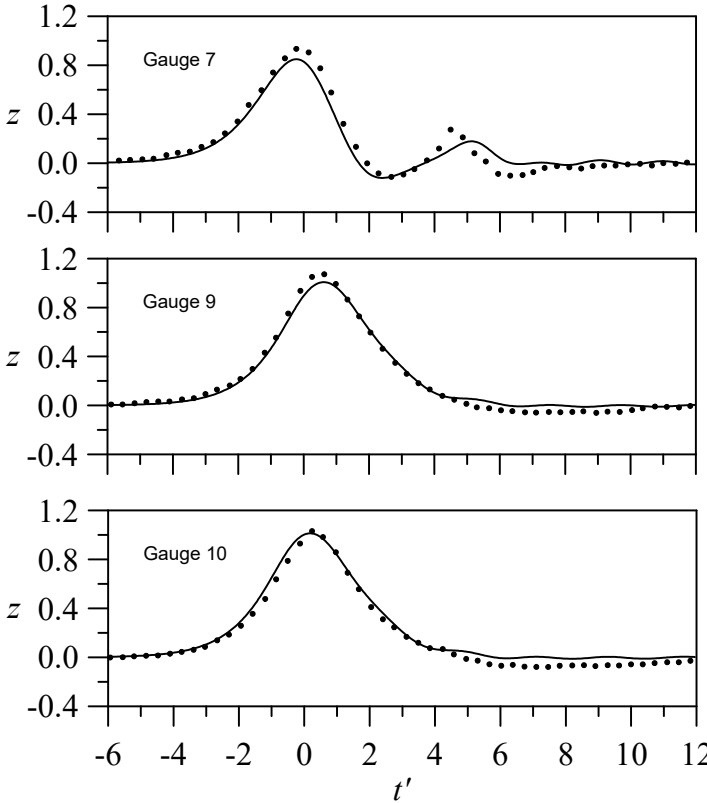

**Figure 3.** Comparisons of wave elevations at six wave-gauge positions.

Next, a single cylinder with a circular structure is considered with a gap between the cylinder bottom and channel bed. Chen and Wang [45] experimentally measured the wave elevations of a solitary wave passing through a cylinder that does not make complete contact with the bottom bed. The undisturbed water depth of their experiment was $H^* = 7.62$ cm, and according to its dimensionless channel experimental conditions, its length was $(-40, 40)$, width was $(-2.5, 2.5)$, wave gauge positions G1 and G2 were $-1.411$ and $1.411$, respectively, and the cylinder diameter $D$ was 1.5. The wave gauge measurements reported by the authors are for the case of $H_s = 0.5$. Compared with Chen and Wang's measured results shown in Figure 4, the results of the numerical simulation obtained in this study, which are consistent with their experimental results, are shown in Figure 4a,b with $A_0 = 0.19$ and Figure 4c,d with $A_0 = 0.31$. Because their channel width was $W = 5$ (only five times the still water depth), the cylinder with $D = 1.5$ was very close to the side wall of the channel. Figure 5 compares the difference in results obtained for $W = 5$ and 60, and shows that the wide channel $W = 60$ causes the scattering dispersive waves around the cylinder to be relatively small and weak.

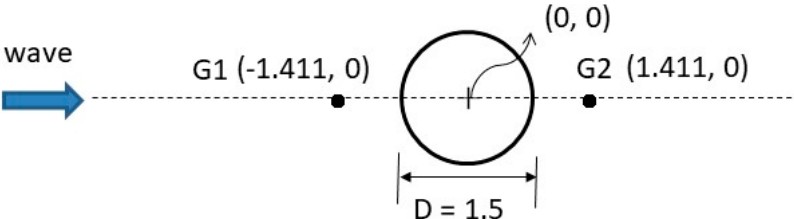

**Figure 4.** *Cont.*

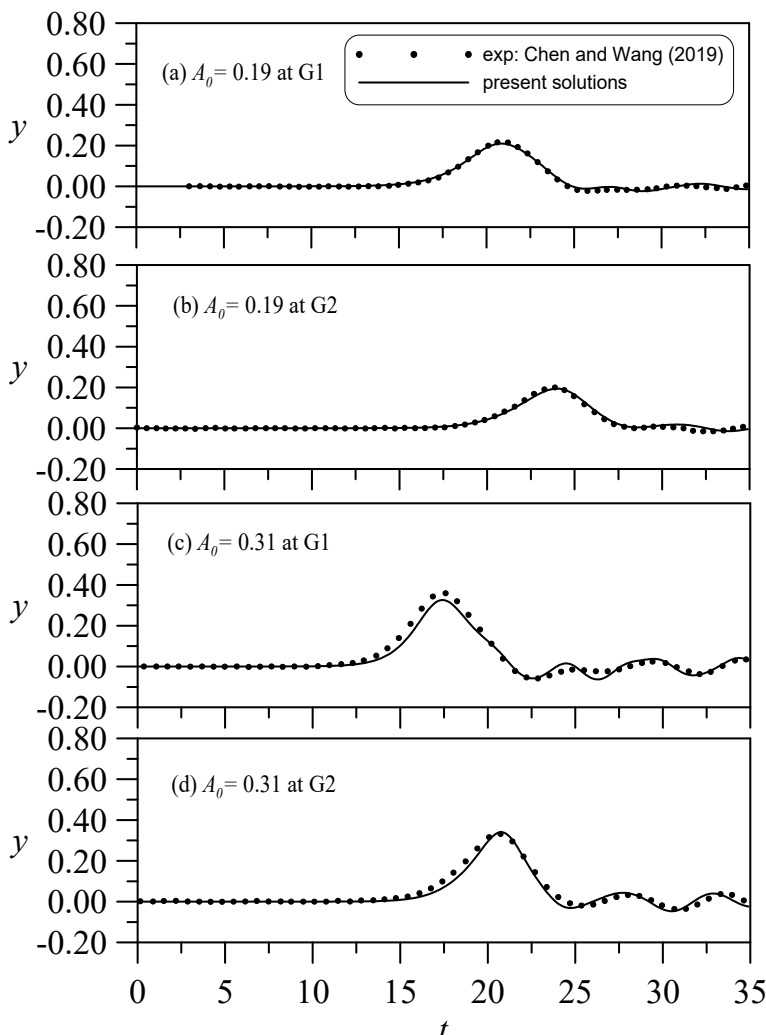

**Figure 4.** Comparisons of wave elevations at two gauges for a solitary wave passing through a non-touching seabed cylinder ($H_s = 0.5$).

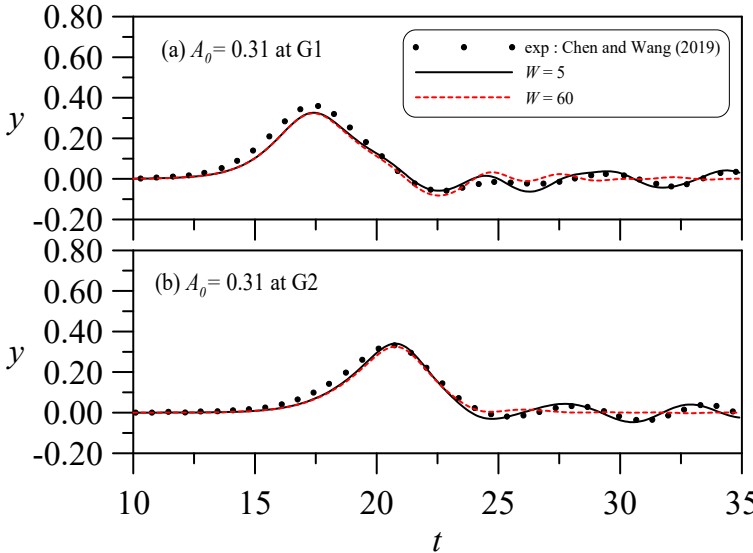

**Figure 5.** Comparisons of wave elevations at two gauges for a solitary wave passing through a non-touching seabed cylinder ($H_s = 0.5$). The results of $W = 5$ and 60 calculated and compared with the experimental condition with $W = 5$.

## 4. Results and Discussion

### 4.1. Solitary Wave Hitting a Circular Cylinder with Different Drafts and Sizes

Here, the influence of the gap between the bottom of the cylinder and the seabed on a solitary wave hitting the cylinder is analyzed. A wave with $A_0 = 0.3$ passes through a cylinder of $D = 4$, for which different immersion depths ($H_s$) are simulated. Figure 6 shows a plot of the wave motions for a circular cylinder that is semi-immersed at $H_s = 0.5$, with the left side of each figure showing the top view to enable observation of the evolutions of the crest lines, and the right side showing a 3D perspective. For example, in Figure 6 at $t = 0$, the distance of a solitary wave from the center of the cylinder 15 times the water depth. In this figure, it can be seen that when $t = 3$, the solitary wave approaches the cylinder but is not significantly affected by it. When $t = 12$, the wave touches the cylinder's surface, and runs up the front of the cylinder. Then, most of the wave passes through the cylinder to form a transmitted wave. The wave is also scattered by the cylinder, and the water elevation initially runs up the front of the cylinder and then drops and forms a system of diffraction waves around it. At $t = 18$, the wave passed through the cylinder, and the crest lines behind the cylinder are just slightly lower than those on both sides, such that the cylinder does not seriously eradicate the crest lines. After the wave has completely passed through the cylinder (such as at $t = 30$), the shape of the crest line that was damaged by the cylindrical structure returns to that of the original incident wave and continue to propagate forward. That is, the crest lines gradually become straight again. Eventually, a system of cylindrical diffraction waves radiates around the cylinder.

When a solitary wave passes through a cylinder, an impact runup is generated in front of the cylinder, and a water level squeeze and rise occurs at the cylinder's rear surface due to diffraction. The rising wave elevation has a great impact on the structure of the cylinder, so it is important to analyze the change in water levels at the front and rear of the cylinder for different submerged depths, $H_s$. Figure 7a shows the water level histories of the front cylinder at point $A$ $(-2, 0)$, and Figure 7b shows those of point $B$ $(2, 0)$ behind the cylinder. This figure shows the differences among $H_s = 1.0$ (complete immersion), 0.7, and 0.5. At an incident wave height of 0.3, although the $H_s$ value changes, there is no obvious difference in the overall phenomenon. The maximum water level at point $A$ can increase to about 0.5, which is about 1.7 times the height of the original wave. Then, the water level at point $A$ drops below the still water level and gradually returns to the still water level. The evolution at point $B$ gradually increases and then decreases, but all the recorded water levels at point $B$ during this interaction are higher than the still water level.

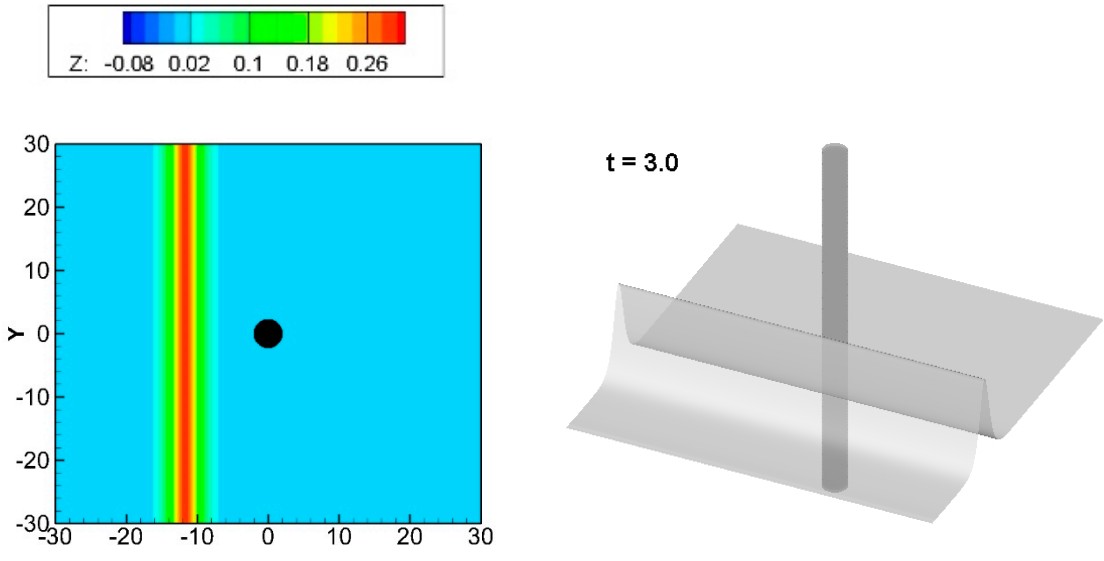

**Figure 6.** *Cont.*

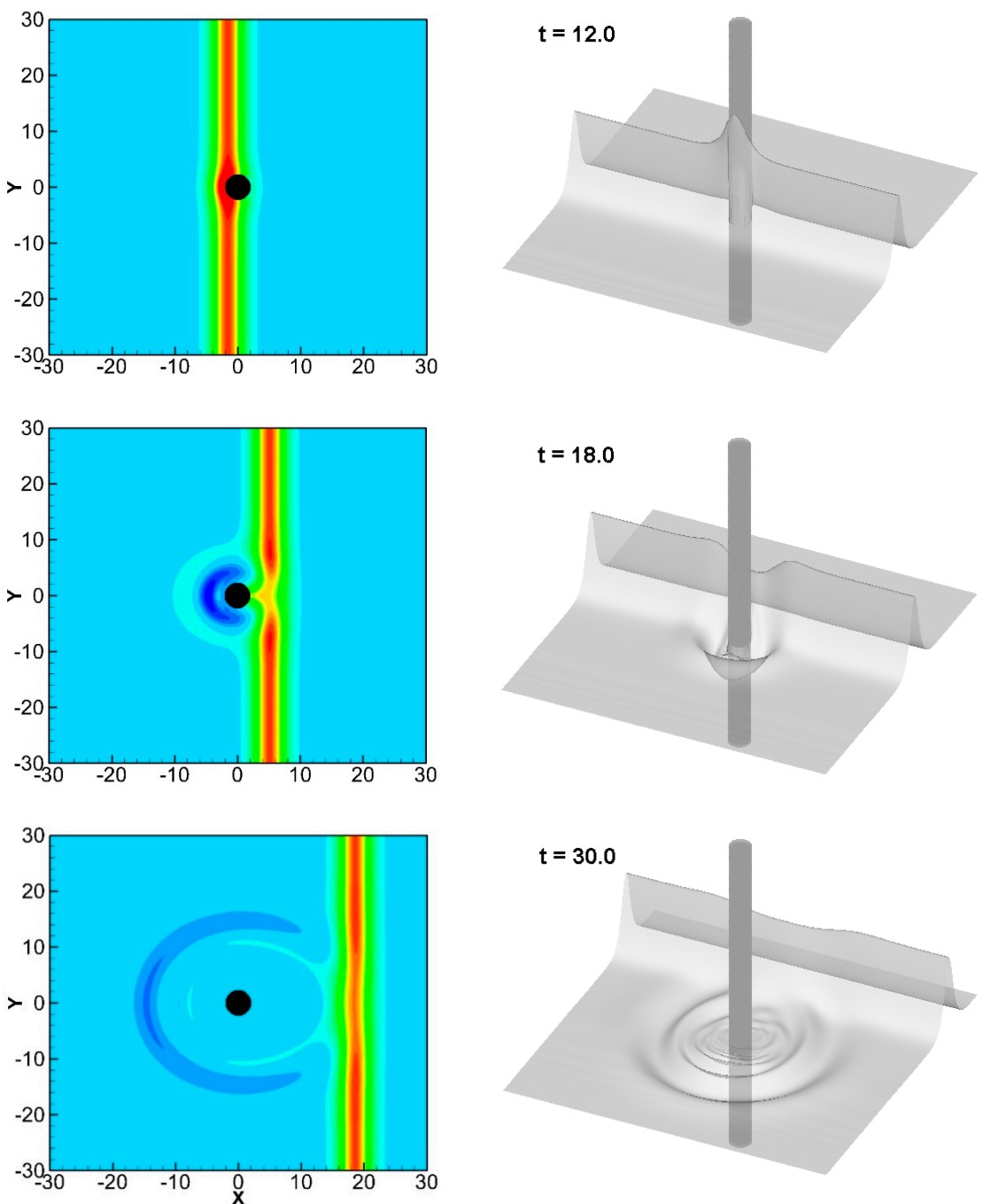

**Figure 6.** Wave patterns of a solitary wave passing through a partially immersed cylinder for $A_0 = 0.3$, $D = 4$, and $H_s = 0.5$. The left is a plane view, and the right is a 3D perspective.

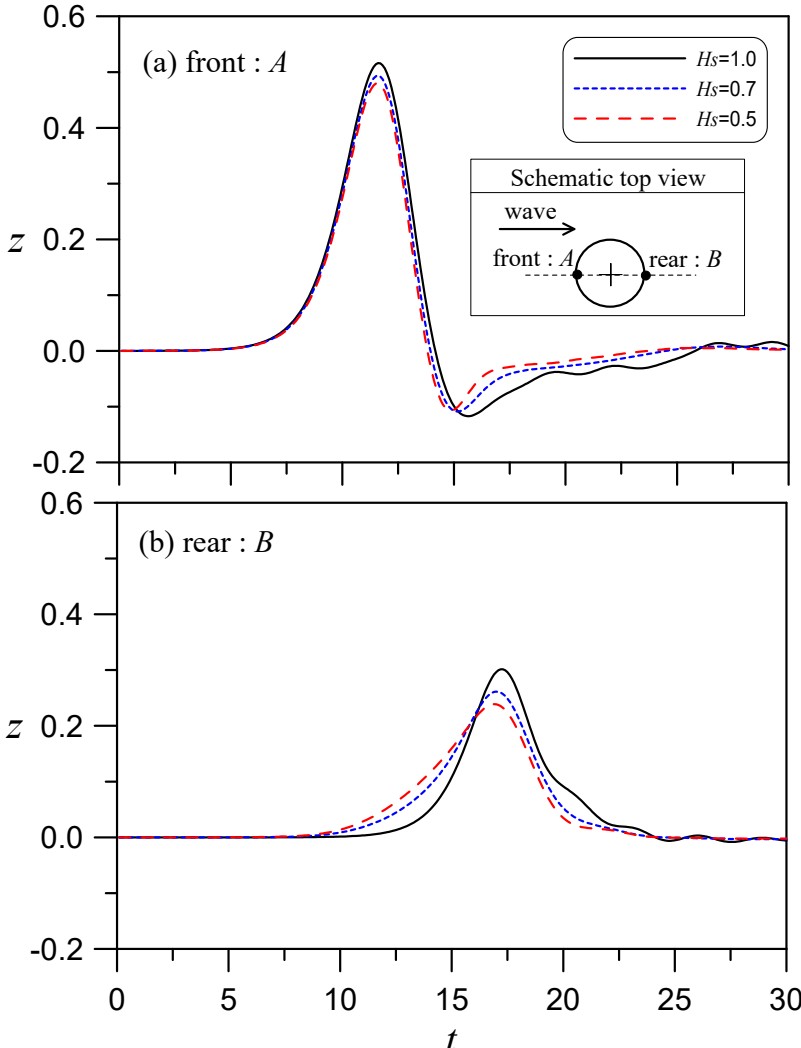

**Figure 7.** Front and rear wave elevations of a solitary wave passing through a vertically immersed circular cylinder in water with different $H_s$ for $A_0 = 0.3$ and $D = 4$.

A careful examination of the effects of $H_s$ for a deep immersion reveals that it produces a large runup and depression at point $A$ and a large increase in the runup at point $B$. When $H_s = 1.0$, the maximum runup elevation of point $B$ nearly reaches the height of the incident wave. However, if $H_s$ is smaller, the runup of point $B$ is slightly reduced, and the water level starts to rise earlier. This means that a larger gap (shallower $H_s$) allows more wave-induced flow to pass under the cylinder to form earlier wave surges at the rear of the cylinder. As such, the water level rise occurs earlier, but the volume of water squeezing and rising from the lateral sides of the cylinder after diffraction is weak, so the maximum runup at point $B$ is low.

Figure 8 shows the vertical-profile differences for a solitary wave passing through a cylinder with immersions $H_s = 1.0$, 0.7, and 0.5. In this figure, the free-surface profiles at the symmetry plane ($y = 0$) are plotted at various moments. When $t = 9$, it can be seen that the wave touches the cylinder, so there is a rising runup at the front of the cylinder. Prior to this time, there is no obvious difference among the three immersions ($H_s = 1.0, 0.7,$ and $0.5$). However, when $t = 12$, the wave rises higher. For example, in the case of $H_s = 0.5$, there is a gap between the bottom of the cylinder and the seabed, so the current caused by the wave will pass through the gap. Therefore, the larger the gap (i.e., shallower $H_s$), the stronger the transmitted wave that initially emerges at the rear of the cylinder. However, the rear water level changes over time. By $t = 15$, a wave reflection and transmission mechanism appears. However, when $t = 18$ and $H_s = 1.0$, complete diffraction occurs (no current moves through the gap), so

more diffraction waves will accumulate from the surrounding lateral areas of the cylinder, which will result in greater surge behind the cylinder, such that the transmitted wave behind the cylinder will become large. That is, at this time, the large gap creates a weaker runup at the rear of the cylinder. At $t = 30$, we can see that the wave bypasses the cylinder and is destroyed but still appears as a solitary wave, even though the wave height is slightly lower than the incident wave height (as denoted by the horizontal dashed line in Figure 8f, which represents the original wave height). A detailed comparison of the cases of $H_s = 0.5$ and $H_s = 1.0$ reveals that the transmission wave height at $H_s = 1.0$ is slightly high and is accompanied by some trailing waves, but the reflected waves produce more trailing waves for the case of $H_s = 0.5$. Generally, there is no significant difference between them.

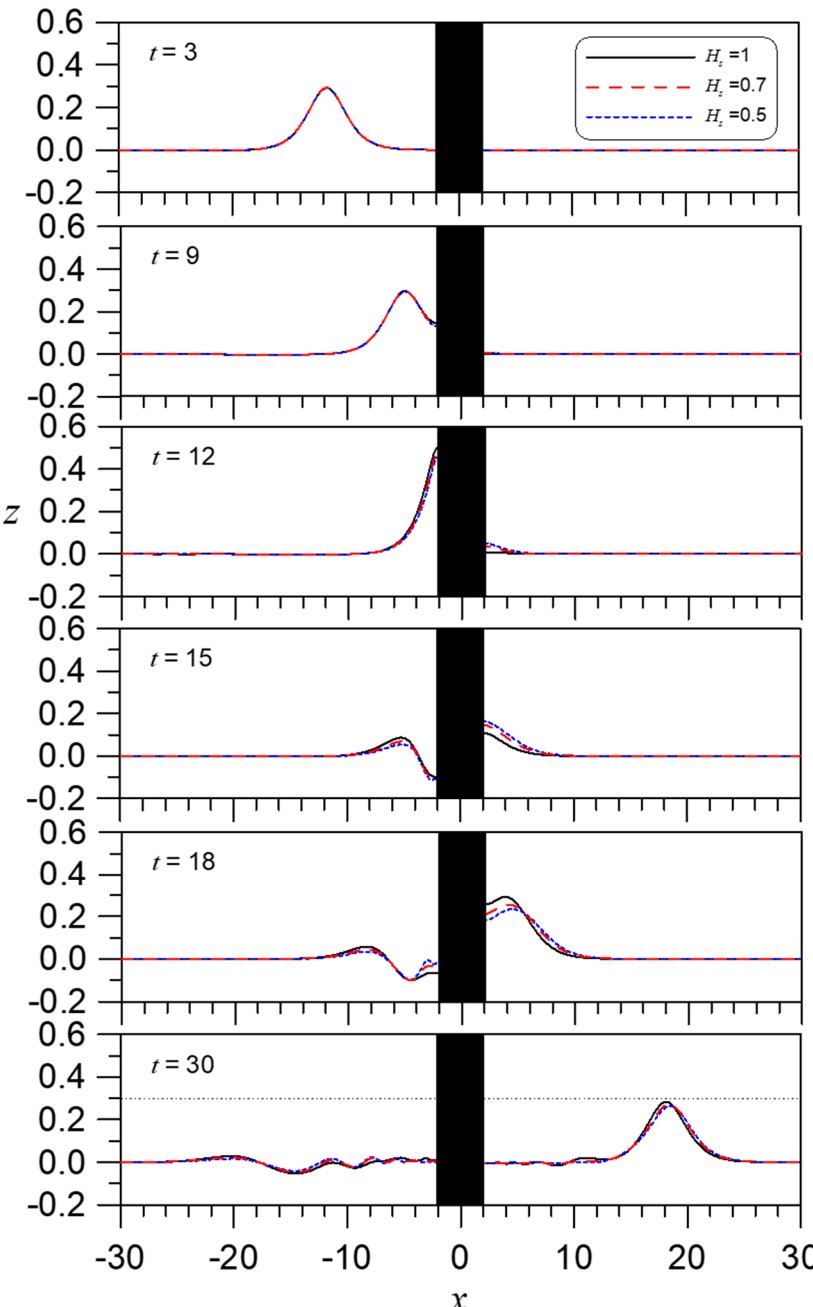

**Figure 8.** The solitary wave $A_0 = 0.3$ passes through an upright circular cylinder with $D = 4$ for $H_s = 1.0$, 0.7, and 0.5, comparing the wave profiles on $y = 0$.

Figure 9 shows a comparison of the impact of different cylinder sizes on the waves at the front and rear of the cylinder. We can see that the larger the cylinder, the larger the front runup and the smaller the rear runup. If $D$ is fixed, as shown in the figure, the larger is $H_s$, the larger are both the front and rear runups. This echoes the results shown in Figure 7.

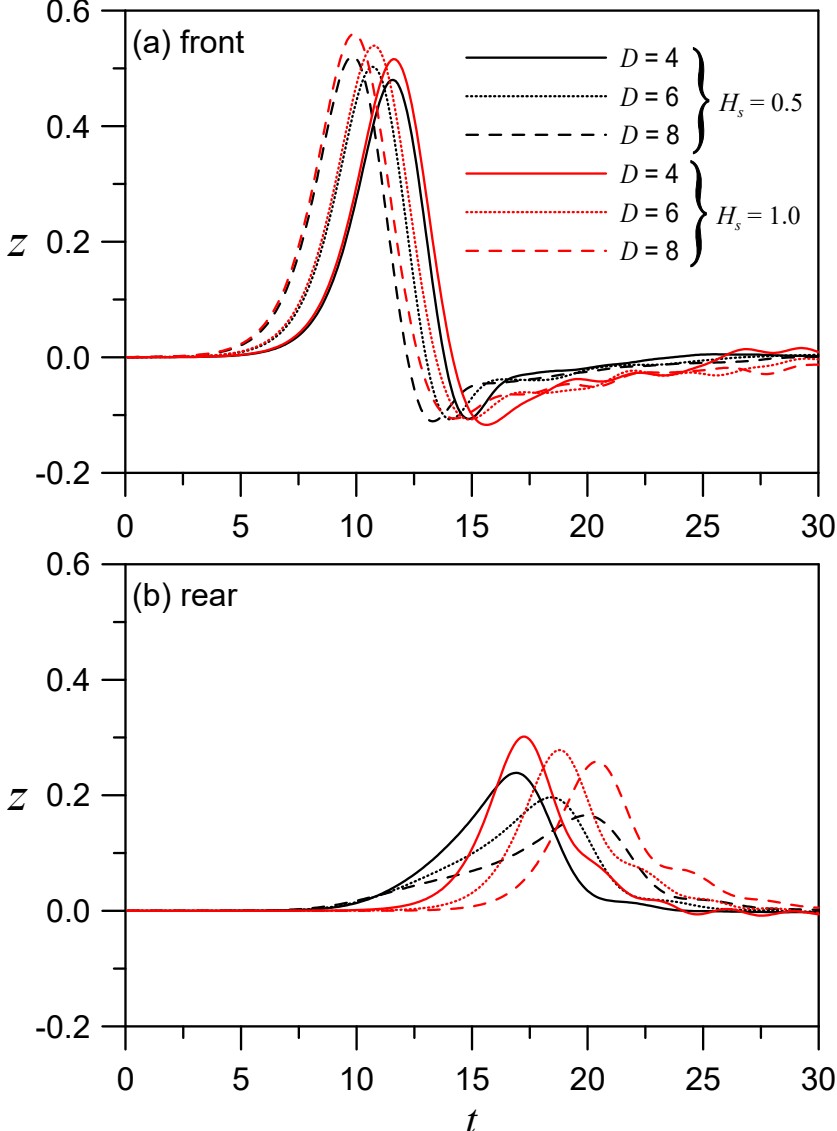

**Figure 9.** Front and rear wave elevations of a solitary wave passing through a vertically immersed circular cylinder in water for $A_0 = 0.3$; $H_s = 0.5$ with different $D$ and $H_s$.

### 4.2. Solitary Wave Hits a Circular Cylinder with a Hollow Zone

A wave passing through a hollow cylinder will generate an oscillation effect in the hollow water column, which is helpful information for the design of an OWC device. This section discusses the problem of solitary waves passing through a hollow cylinder (Figure 10). In addition to the incident wave height ($A_0$), other possible influencing parameters are the still water depth $H$ ($H = 1$ after normalization), the cylindrical immersion depth $H_s$, and the outer diameter $r_1$ and inner diameter $r_2$ of the concentric circular cylinder (cylinder thickness $dr = r_1 - r_2$).

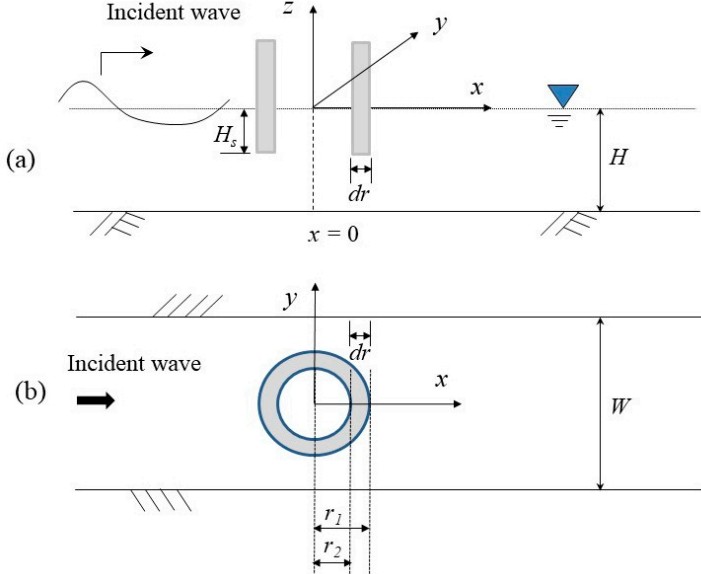

**Figure 10.** Schematic diagram of waves passing through a hollow cylinder: (**a**) side view on $y = 0$ and (**b**) top view.

For simplicity, this problem is discussed using $A_0 = 0.3$ as an example, mainly to analyze the influence of $r_1$, $dr$, $H_s$, and $W$ on wave oscillation in the hollow circular cylinder. Figure 11 shows a hollow cylinder with $r_1 = 2$, $r_2 = 1$ ($dr = 1$), immersed at $H_s = 0.5$, with a channel width $W = 10$ to determine the characteristics of the change in the surrounding water level when a solitary wave passes through this cylinder. Figure 11a shows plots of the time history of the water level at three positions, a, b, and c, in the transverse direction. It is apparent that the wave runup and rundown at points a and c are similar to those obtained in the previous water-level analysis of a wave passing through a single solid cylinder at its front and rear positions, whereas Point b shows a change in the water-level oscillation in the hollow area, with its oscillation amplitude obviously larger than those of Points a and c. Figure 11b shows the water-level changes at Points e and f in the longitudinal direction. This figure shows that Point f in the hollow area has a significant oscillation amplitude similar to that of Point b, which indicates that the hollow area causes waves to generate an oscillation effect in the hollow column area.

To analyze the influence of $dr$ (or $r_2$), Figure 12 shows the case of $r_1 = 3$, $H_s = 0.5$, and $W = 20$. That is, the outer diameter is held constant while the thickness (or inner diameter) is varied to observe how the water oscillates in the hollow column. A comparison of Figure 12a,b reveals that in Figure 12a, when $r_2 = 1$, the difference in the water levels at points a, b, and c of the hollow area is small. However, in Figure 12b, when $r_2 = 2$ (which is larger than that in Figure 12a), the water levels of the hollow area are significantly different at points a', b', and c'. This result indicates that when $r_2 = 1$ (Figure 12a), the water level of the hollow area oscillates more uniformly. Thus, if the hollow area is small (not greater than the still water depth), the hollow water column will fluctuate more uniformly. This phenomenon can be conceptualized and anticipated and can also be observed in the 2D water-level color contour map in Figure 13. Figures 13a–g and 13a'–g' correspond to Figure 12a,b, respectively. The planar view shows the overall changes in the reflection, transmission, and diffraction of the wave as it encounters the hollow cylinder. This result is indicated by the color change in the hollow zone. The hollow area in Figure 13a–g is small ($r_2 = 1$) and always shows a single color in subsequent figures, which indicates that its water level is uniform in the hollow zone. In contrast, the color of the hollow area in Figure 13a'–g' is not uniform.

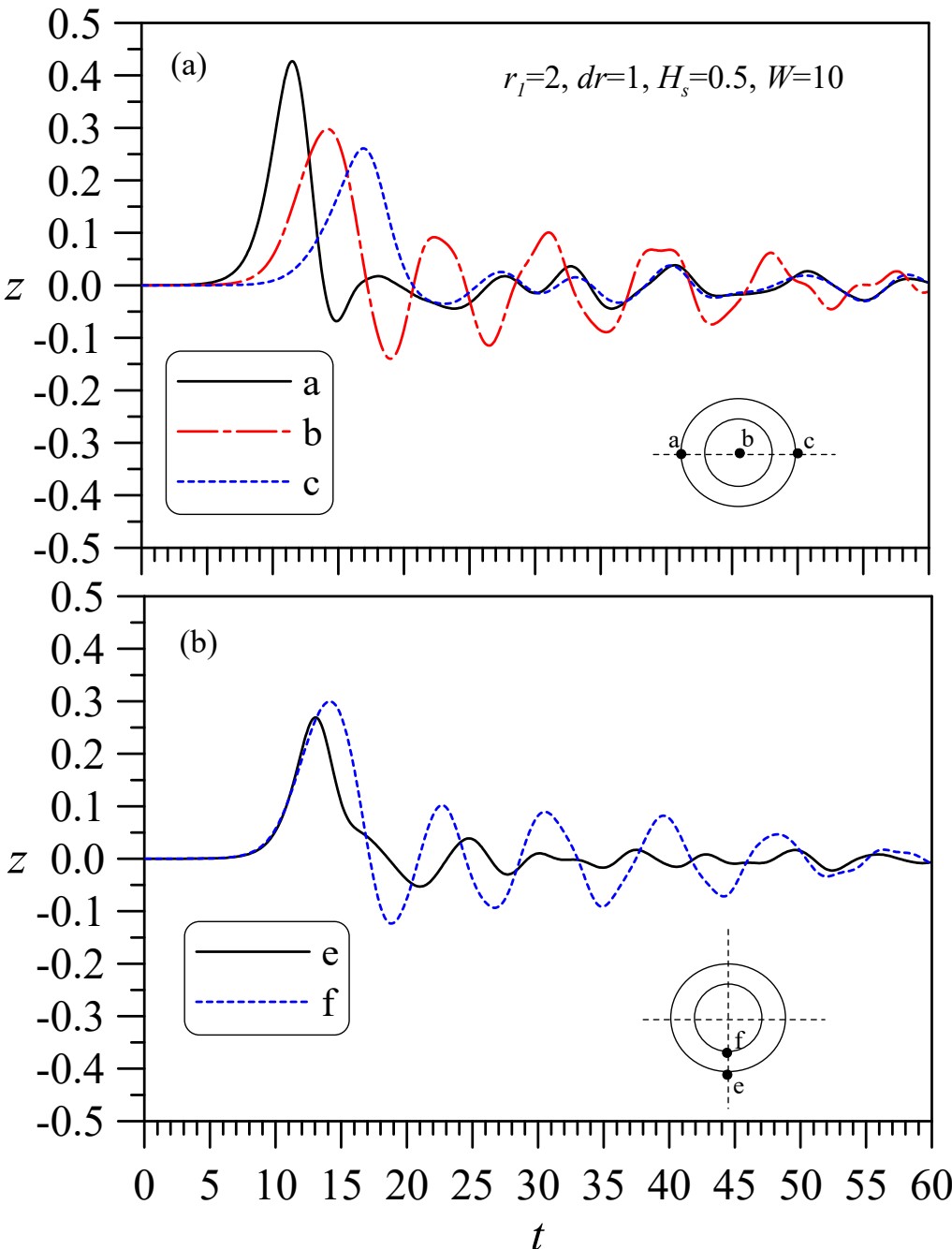

**Figure 11.** A solitary wave of height $A_0$ = 0.3 passes through a hollow circular cylinder generating the time histories of the water level at surrounding points.

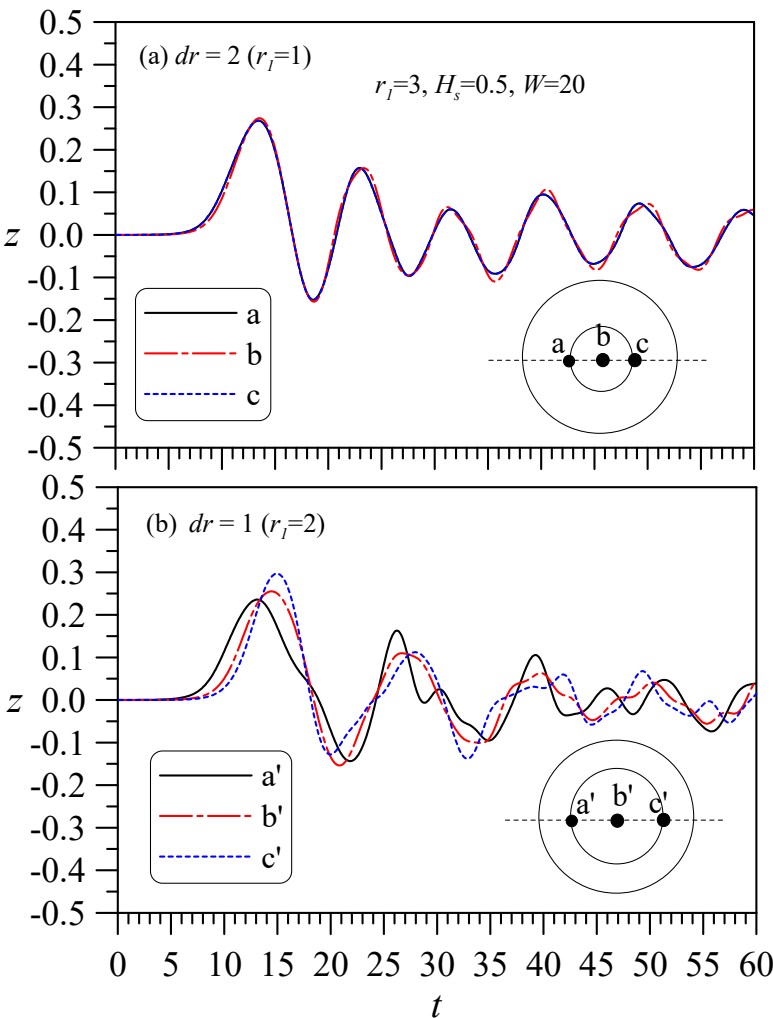

**Figure 12.** A solitary wave of height $A_0 = 0.3$ passes through the hollow circular cylinder recording wave elevations at points in the hollow area. A comparison of the wave elevations in the hollow area of the hollow cylinder with different inner diameters.

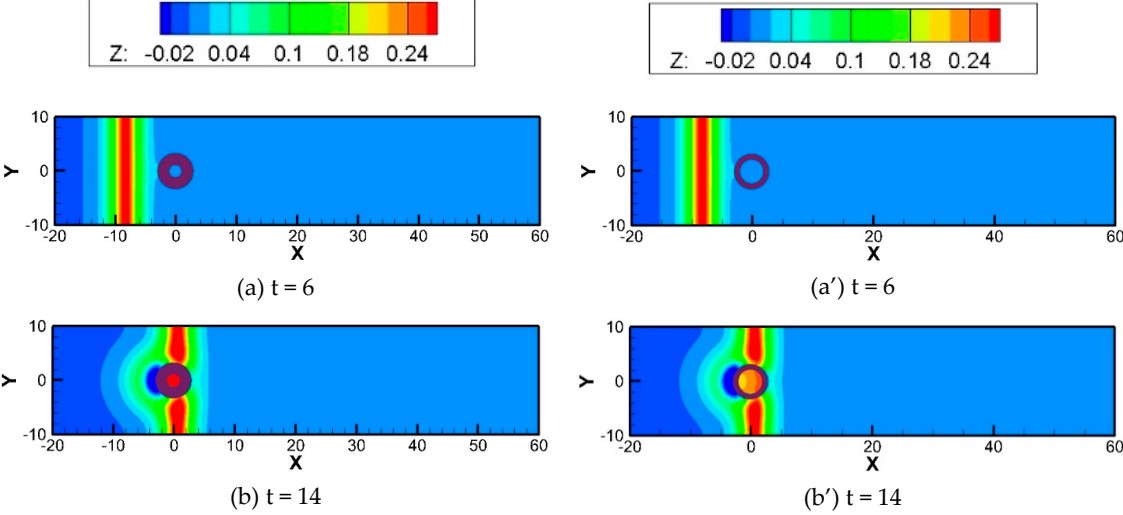

**Figure 13.** *Cont.*

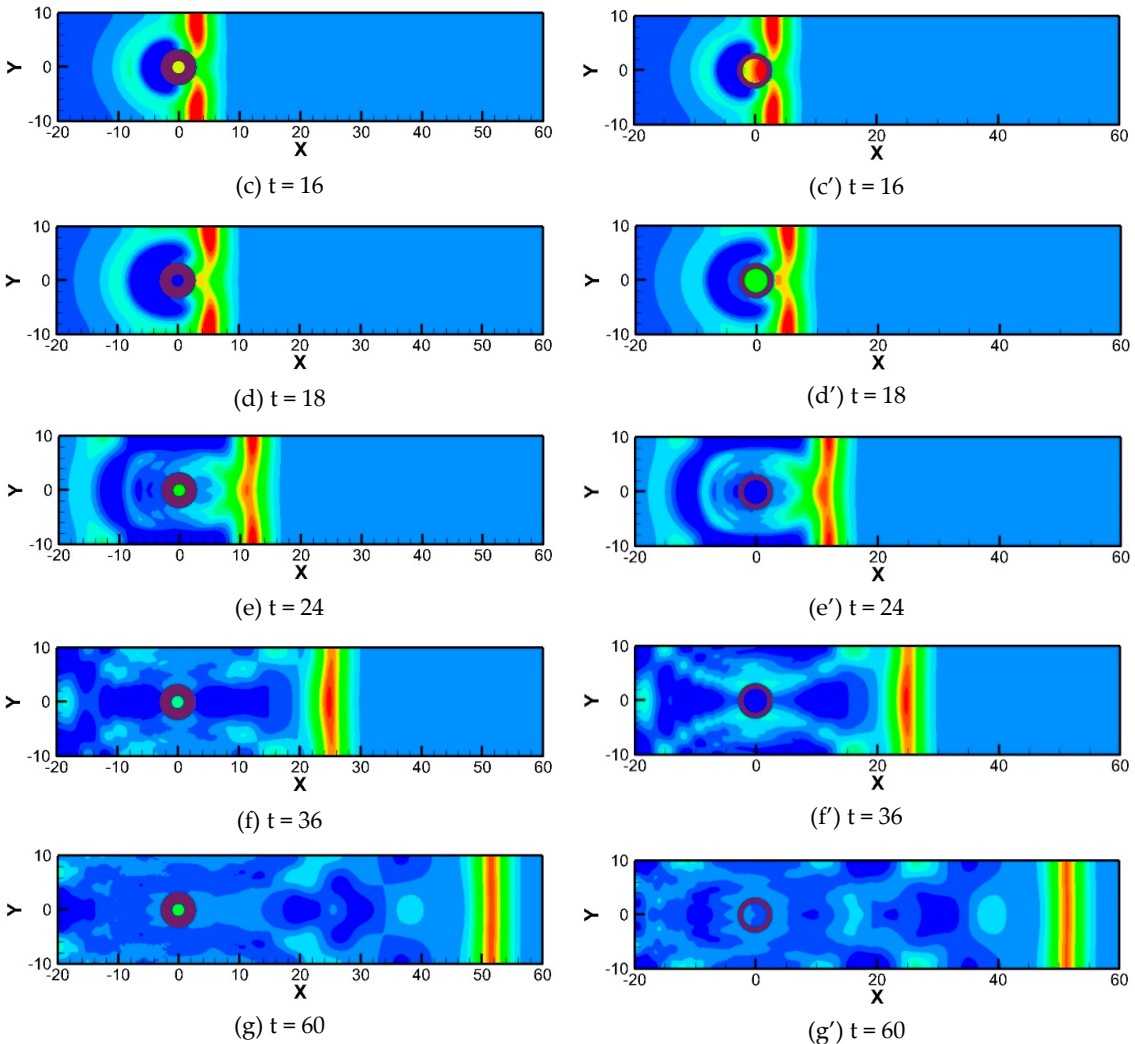

**Figure 13.** The wave-elevation contours at various times for a solitary wave $A_0 = 0.3$ propagating in the channel width $W = 20$ through the hollow cylinder with immersion depth $H_s = 0.5$. The cylinder's conditions are (**a–g**): $r_1 = 3$, $r_2 = 1$ and (**a'–g'**): $r_1 = 3$, $r_2 = 2$.

Next, the influence of the outer diameter, inner diameter, and cylinder thickness on the fluctuation is considered when $W$ and one of the other factors are fixed.

### 4.2.1. Fixed Thickness, with Changes in the Outer and Inner Diameters

Figure 14 shows the water levels for a fixed channel width $W = 15$ and thickness $dr = 1$, with the outer diameters $r_1 = 2, 3$, and 4. Figure 8a–c show the effect of different $H_s$ values on the rise and fall of the water column. These figures reveal that when $dr$ is fixed and the outer diameter becomes larger, there will be a larger hollow area, which is not conducive to the formation of a uniform water level over the entire hollow area. A smaller $r_1$ value results in a better oscillation effect, and the larger is $H_s$ (e.g., $H_s = 0.7$), the more significant the oscillation, i.e., a water column with a smaller $r_1$ and a larger $H_s$ has a better oscillation effect.

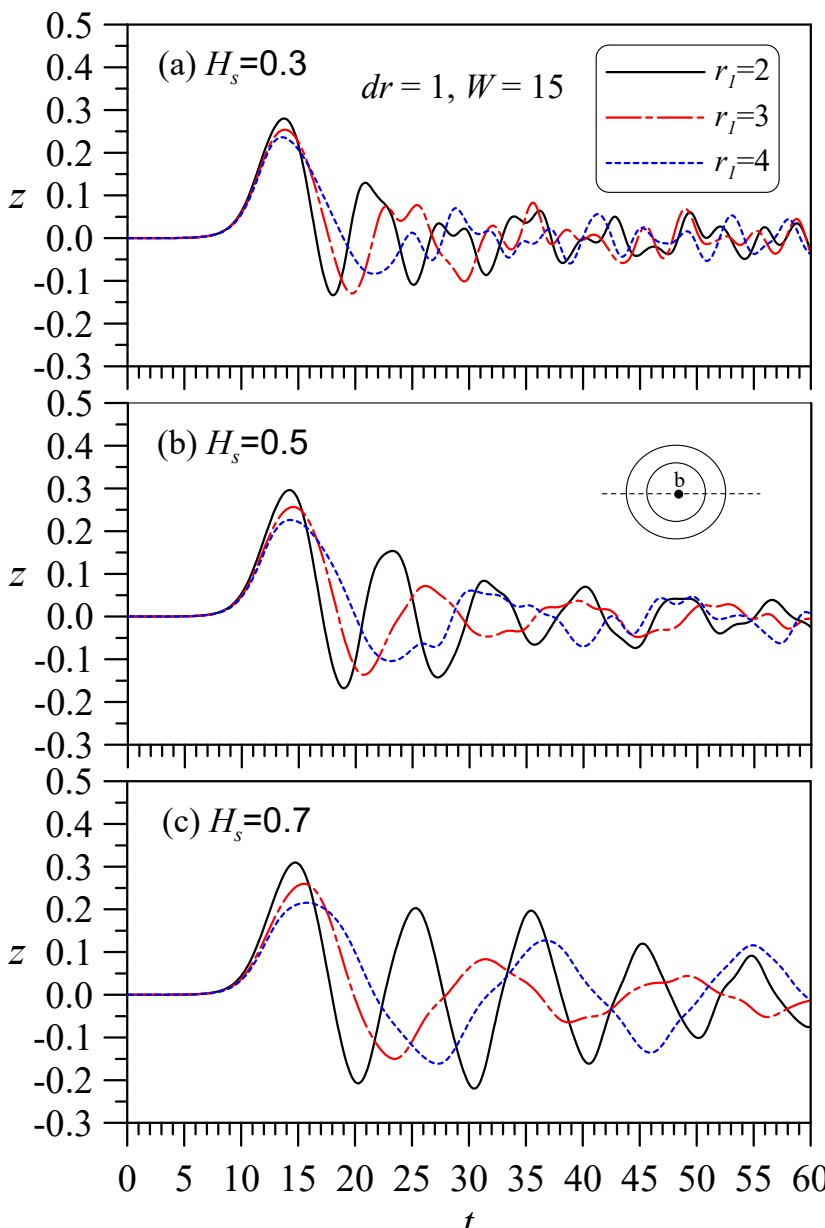

**Figure 14.** Influence of outer diameter (fixed wall thickness) is considered: A solitary wave of height $A_0 = 0.3$ passing through a hollow cylinder has the same cylindrical plate thickness ($dr = 1$) but different outer diameters ($r_1$). The water level of the hollow center point varies with time. The immersion depths ($H_s$) are not equal: (**a**) $H_s = 0.3$, (**b**) $H_s = 0.5$, and (**c**) $H_s = 0.7$.

### 4.2.2. Fixed Inner Diameter, with Changes in the Outer Diameter and Thickness

Another case for analyzing the influence of the outer diameter is obtained by fixing the inner diameter $r_2$ but changing the thickness. Similar to the analysis shown in Figure 14, Figure 15 shows the water levels for a fixed inner diameter $r_2 = 1$ and channel width $W = 15$. The inner diameter of the hollow area is fixed so the influence of the outer diameter $r_1$ can be observed. Figure 15a shows that when the cylinder is not deeply immersed (e.g., $H_s = 0.3$ in Figure 15a), a larger outer diameter helps to drive the amplitude to generate more regular motions. In Figure 15a, for the case of $H_s = 0.3$, the larger is $r_1$, the larger the amplitude (except for the main wave). If the immersion depth is deep, this phenomenon does not occur, but a smaller outer diameter brings about larger amplitudes. That is, if the immersion depth is not deep, the outer diameter can greatly improve the amplitude height and regularity, but a deeper immersion depth and smaller outer diameter can generate larger amplitudes.

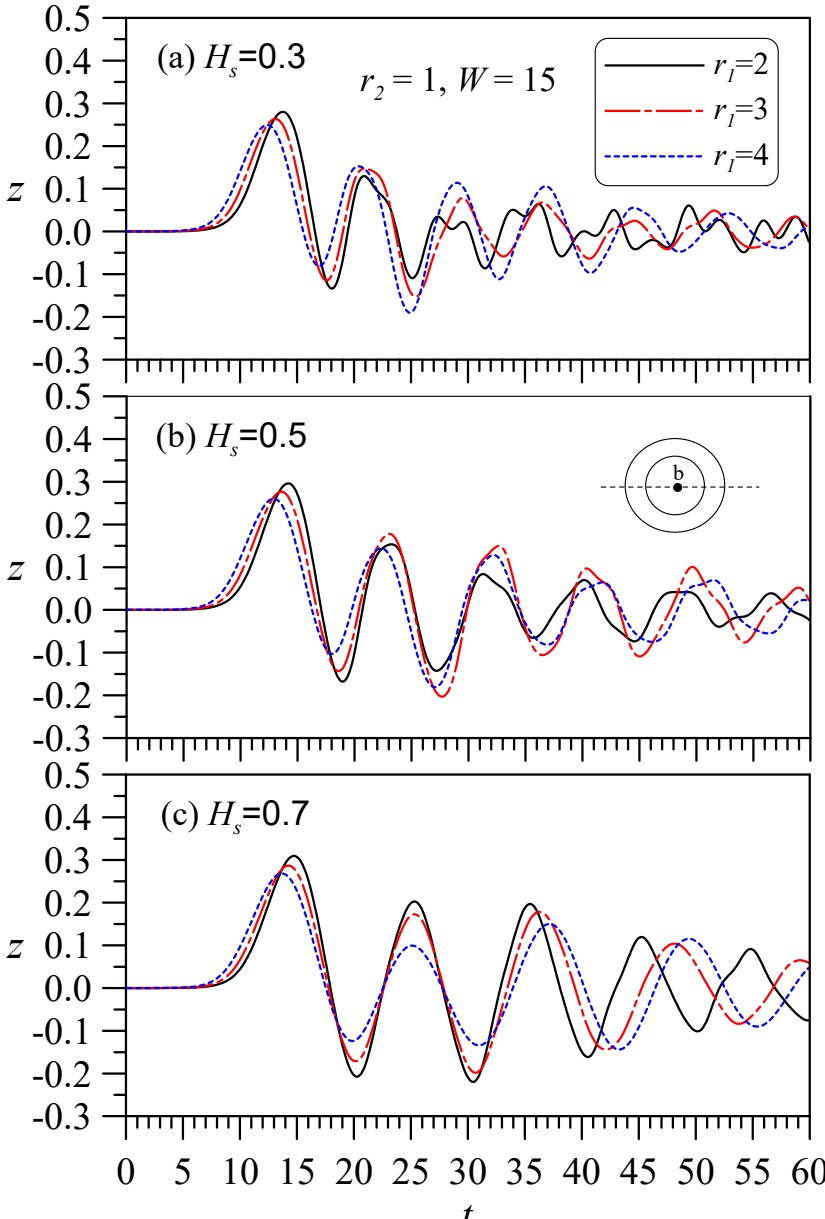

**Figure 15.** The influence of outer diameter $r_1$ (fixed $r_2$) is considered: A solitary wave height $A_0 = 0.3$ passes through a hollow cylinder with the same inner diameter ($r_2 = 1$) but different outer diameters ($r_1$) and the water level at the central point of the hollow is recorded over time. The immersion depths ($H_s$) are not equal: (**a**) $H_s = 0.3$, (**b**) $H_s = 0.5$, and (**c**) $H_s = 0.7$.

### 4.2.3. Fixed Outer Diameter, with Changes in the Inner Diameter and Thickness

For a fixed outer diameter $r_1 = 4$ and channel width $W = 20$, here, the effect of changing the thickness is analyzed. Figure 16 shows the case of a fixed outer diameter, which reveals that the greater the thickness (that is, the smaller the hollow area), the greater the oscillation, and if $H_s$ is larger (e.g., Figure 16c), this phenomenon will be more obvious. However, the greater the thickness of the structure, the greater the manufacturing cost, which is not practical. Therefore, it is also recommended that the outer diameter not be too large.

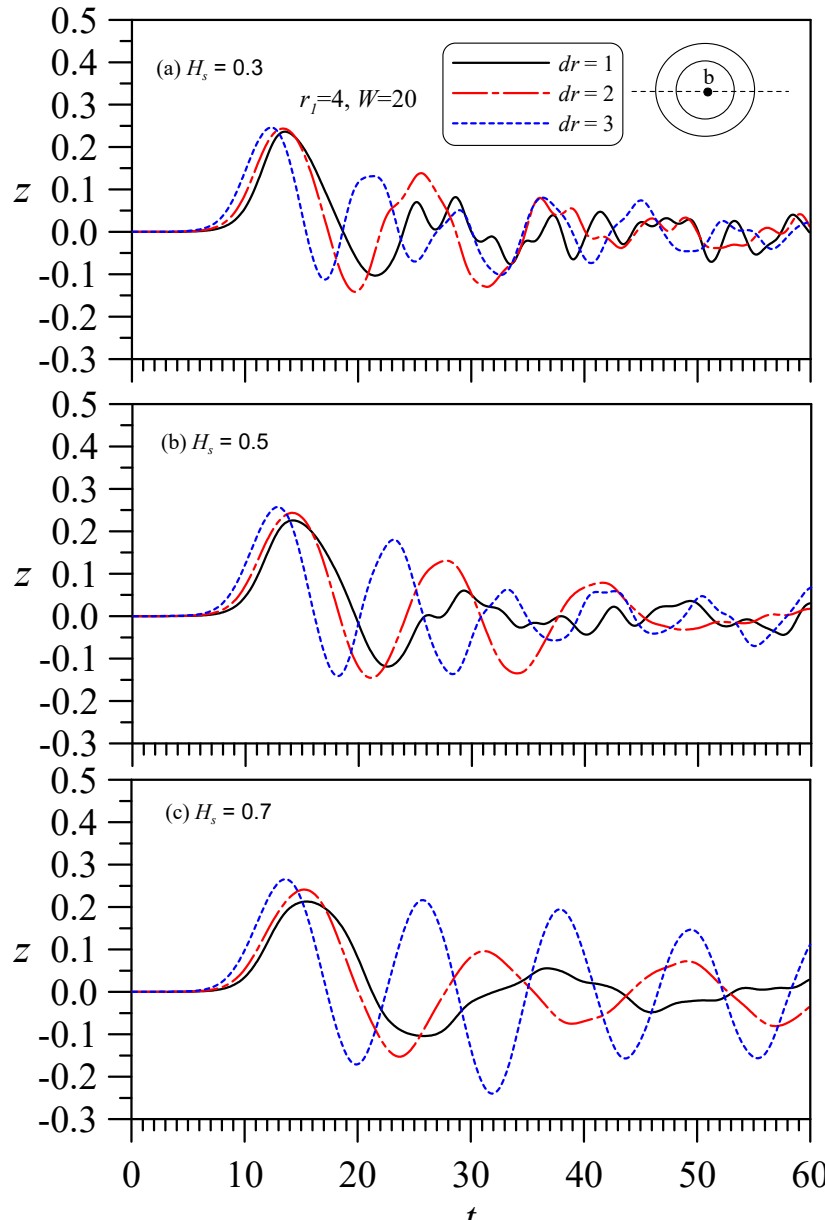

**Figure 16.** The influence of cylinder thickness ($dr$) is considered: A solitary wave of height $A_0 = 0.3$ passes through the hollow cylinder with the same outer diameter ($r_1 = 4$) but different cylinder plate thicknesses ($dr$). The water levels at the hollow center point are recorded over time. The immersion depths ($H_s$) are not equal: (**a**) $H_s = 0.3$, (**b**) $H_s = 0.5$, and (**c**) $H_s = 0.7$.

### 4.3. Influence of W on the Solitary Wave Hitting a Circular Cylinder with a Hollow Zone

Earlier in this article, the effect of $W$ on a solitary wave against a solid cylinder was considered (Figure 5). The results indicated that a wide response channel smoothens the fluctuation in the diffraction of a solitary wave through a cylinder. In this subsection, the influence of $W$ on a hollow cylinder is considered. If $r_1 = 2$ and $r_2 = 1$ ($dr = 1$) are fixed, the influence of $W$ can be analyzed. Figure 17 shows that when $H_s = 0.3$ or $0.5$, there is no obvious influence by $W$, but when $H_s = 0.7$, $W = 15$ produces larger oscillations than either $W = 10$ or $20$. This result is somewhat confusing. To understand why $W = 15$ produces larger amplitudes, other $W$ values were analyzed ($W = 10–20$) for the case of $H_s = 0.7$. Figure 18 shows that there are larger amplitudes when $W = 16$. Under the conditions of a fixed hollow cylinder size and immersion depth, there is an optimal $W$ value that generates the maximum amplitude effect.

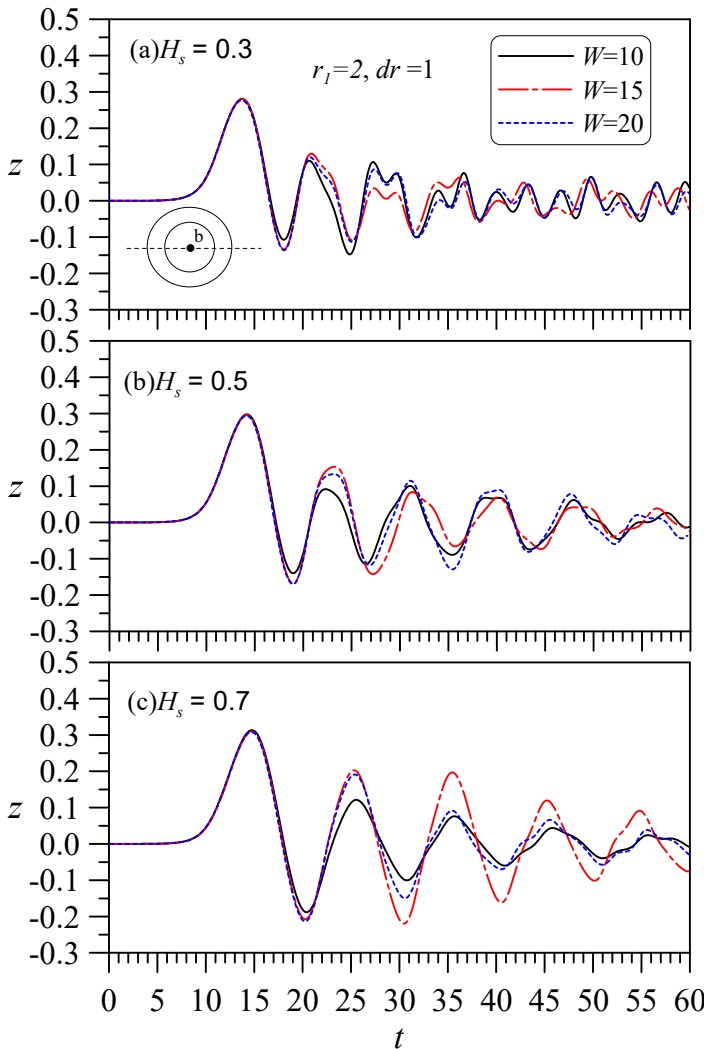

**Figure 17.** The influence of channel width $W$ is considered: A solitary wave of height $A_0 = 0.3$ passes through a hollow cylinder with the same outer diameter ($r_1 = 2$) and inner diameter ($r_1 = 1$) but different channel widths ($W$). Observe the wave elevations at the center point varying with time. The immersion depths ($H_s$) are not equal: (**a**) $H_s = 0.3$, (**b**) $H_s = 0.5$, and (**c**) $H_s = 0.7$.

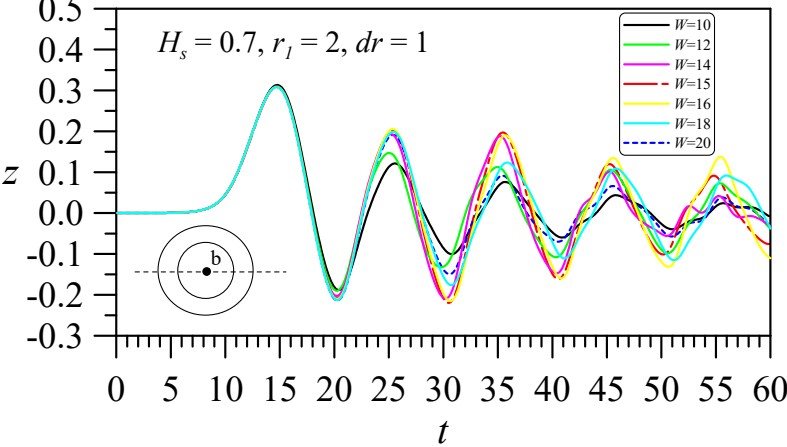

**Figure 18.** Comparisons of time histories of wave elevations at the central point for a solitary wave with $A_0 = 0.3$ passing through a hollow cylinder with the same outer diameter ($r_1 = 2$), inner diameter ($r_1 = 1$), and immersion depth $H_s = 0.7$ but different channel widths ($W$).

## 5. Conclusions

In this study, a 3D fully nonlinear potential wave model was applied to consider the interaction of a solitary wave with a vertical cylinder piercing the water at different depths with or without a hollow zone. A solitary wave with $A_0 = 0.3$ was used as the typical incident condition, and the model was first compared with the experimental data obtained by other researchers for verification, and was then applied to explore the wave–cylinder interactions. The conclusions are summarized below.

We first considered the interaction of a solitary wave with a single cylinder that has full contact with the seabed and obtained reasonable results. For a solitary wave $A_0 = 0.3$ passing through a cylinder of $D = 4$, the peak of the wave passing through the cylinder was determined to be slightly lower than the peaks on both sides that are undisturbed by the cylinder. When the wave has completely passed the cylinder, the waveform destroyed by the cylinder can nearly be restored. The observed runup elevations of the cylinder reveal that the maximum water level in front of the cylinder is about 0.5, which increases to approximately 1.7 times the original wave height (0.3) and then drops and gradually returns to the still water level. The runup evolutions behind the cylinder gradually increase and then decrease, but the recorded water elevations at the front and rear points remained higher than the still water level. This phenomenon changes when there is a gap between the bottom of the cylinder and the seabed. We analyzed the effect of the size of the gap between the bottom of a single cylinder and the seabed on the runup and rundown evolutions. Overall, the main effect of the immersion depth $H_s$ is that when the $H_s$ value is deep, greater runup and rundown occur in front of the cylinder, and the maximum water elevation behind the cylinder is larger. However, the water level behind the cylinder is always higher than the still water level. Regarding the effect of cylinder size $D$, the simulation results indicate that the larger the $D$ value, the larger the front runup and the smaller the rear runup.

We also considered the oscillation effect of solitary waves passing through a hollow cylinder. The influencing parameters include $A_0$, $r_1$, $r_2$ ($dr$), $H_s$, and $W$, for which $A_0 = 0.3$; $r_1 = \sim 1$–4; $r_2 = \sim 1$–3; $H_s = 0.3$, 0.5, and 0.7; and $W = 10$, 15, and 20 were considered to analyze the oscillation in the hollow area under these conditions. The analysis showed that if $r_2$ is small (e.g., $r_2 = 1$), the hollow area obtains more uniform water level oscillation in the water column; the smaller the outer diameter ($r_1$), the better the oscillation effect; and the deeper the immersion depth (i.e., larger $H_s$), the more significant is this phenomenon. For long waves, when $H_s$ is large and $r_1$ and $r_2$ are small ($r_1 > r_2$), large wave fluctuations can be produced in the hollow area. For the conditions considered in this study, when $r_1 = 2$, $r_2 = 1$, and $W = 16$, and the immersion depth is deeper, more uniform, and larger amplitude oscillation waves were observed in the hollow area. Therefore, the inner diameter should not exceed the depth of the water, and the outer diameter should not be too large (if the thickness can withstand the external forces, a less thick wall should be used). However, the effect of the $W$ value was found to become more obvious when only $H_s$ is larger. All results show that the larger is $H_s$, the larger the oscillating wave generated in the hollow area.

For nonlinear long waves, the wave energy is almost evenly distributed along the vertical depth, so a hollow cylinder can be more deeply immersed to block and squeeze more wave energy passing through the gap between the seabed and the cylinder bottom as it enters the hollow zone. For a deeper immersion, the vortex effect caused by the flow separation becomes relatively important. This is a problem that the model used in this study could not handle, and will be addressed by a more complete model in future work.

**Author Contributions:** The author (C.-H.C.) has reviewed and organized the literature, and has applied his numerical model to simulate cases, followed by plotting and presenting an analysis of the results. The author has read and agreed to the published version of the manuscript.

**Funding:** This research was funded by the Ministry of Science and Technology, Taiwan, Republic of China (Grant No. MOST 108-2221-E-275-002) and the APC for publishing this paper was also funded by the Ministry of Science and Technology, Taiwan, Republic of China (Grant No. MOST 108-2221-E-275-002).

**Acknowledgments:** This work was supported financially by the Ministry of Science and Technology, Taiwan, Republic of China (Serial No. MOST 108-2221-E-275-002).

**Conflicts of Interest:** The author declares no conflict of interest.

## Appendix A

The following describes the grid generation, equation transformation, and numerical discretization methods.

### Appendix A.1. Grid Generations

The numerical grids are established by curvilinear coordinates. An algebraic grid generation technique is used to generate the grids. The grid node numbers arranged along the $\xi$, $\eta$, and $\gamma$ coordinates are $i = 1 \sim IM$, $j = 1 \sim JM$, and $k = 1 \sim KM$. The range of the calculation area in the $x$, $y$, and $z$ directions is assumed to be $x(1)$ to $x(IM)$, $y(1)$ to $y(JM)$ and $z(i, j, 1)$ to $z(i, j, KM)$. In this way, the grid distribution formula is as follows:

$$x = x(i) = x(1) + \Delta x(i-1), \ i = 1 \sim IM \tag{A1}$$

$$y = y(j) = y(1) + \Delta y(j-1), \ j = 1 \sim JM \tag{A2}$$

$$z = z(i, j, k) = z(i, j, 1) + \{Z_{KC} - z(i, j, 1)\}(k-1)/(KC-1), \ k = 1 \sim KC \tag{A3}$$

$$z = z(i, j, k) = Z_{KC} + \{z(i, j, KM) - Z_{KC}\}(k-KC)/(KM-KC), \ k = KC \sim KM \tag{A4}$$

here $\Delta x = \{x(IM) - x(1)\}/(IM-1)$, $\Delta y = \{y(JM) - y(1)\}/(JM-1)$, and $Z_{KC}$ is the vertical position of the bottom of the cylinder (see Figure 1), and $z(i, j, KM)$ is equal to free-surface elevation, that is $\zeta_{i,j}$.

### Appendix A.2. Equation Transformation

The calculation process needs to convert all the equations of the physical domain $(x, y, z; t)$ into the equations of the computational domain $(\xi, \eta, \gamma; \tau)$. After transformation, Equation (1) can be expressed as:

$$g^{11}\phi_{\xi\xi} + g^{22}\phi_{\eta\eta} + g^{33}\phi_{\gamma\gamma} + 2g^{12}\phi_{\xi\eta} + 2g^{13}\phi_{\xi\gamma} + 2g^{23}\phi_{\eta\gamma} + f^1\phi_\xi + f^2\phi_\eta + f^3\phi_\gamma = 0 \tag{A5}$$

where $g^{ij}$ $(i, j = 1, 2, 3)$ and $f^i$ $(i = 1, 2, 3)$ are grid geometric coefficients. Since $x$ and $y$ are uniform grids, and the gridlines in the $z$-direction are vertical and straight, the grid geometric coefficients in the formula can be simplified as:

$$g^{11} = 1/x_\xi^2,$$

$$g^{22} = 1/y_\eta^2,$$

$$g^{33} = [(z_\xi y_\eta)^2 + (y_\eta x_\xi)^2 + (x_\xi z_\eta)^2]/J^2,$$

$$g^{13} = -z_\xi/(x_\xi^2 z_\gamma),$$

$$g^{12} = 0,$$

$$g^{23} = -z_\eta/(y_\eta^2 z_\gamma),$$

$$f^1 = f^2 = 0,$$

and $f^3 = -\left(g^{11}z_{\xi\xi} + g^{33}z_{\gamma\gamma} + g^{22}z_{\eta\eta} + g^{13}z_{\xi\gamma} + g^{23}z_{\gamma\eta}\right)/z_\gamma$
where $J = x_\xi y_\eta z_\gamma$.

The free-surface boundary conditions of Equations (2) and (3) can be expressed as:

$$\zeta_\tau \ = \ w - u(\zeta_\xi/x_\xi) - v(\zeta_\eta/y_\eta) \tag{A6}$$

$$\phi_\tau - w\zeta_\tau + \frac{1}{2}(u^2 + v^2 + w^2) + \zeta \ = \ 0 \tag{A7}$$

In Equations (12) and (13), the water particle velocity ($u$, $v$, $w$) can be expressed as:

$$u \ = \ \phi_x \ = \ (\phi_\xi z_\gamma y_\eta - \phi_\gamma z_\xi y_\eta)/J \tag{A8}$$

$$v \ = \ \phi_y \ = \ (-\phi_\gamma x_\xi z_\eta + \phi_\eta x_\xi z_\gamma)/J \tag{A9}$$

$$w \ = \ \phi_z \ = \ \phi_\gamma y_\eta x_\xi/J \tag{A10}$$

The lateral conditions of Equation (4) can be derived for $\phi$ and $\zeta$, respectively, as:

$$\phi_\tau - w\zeta_\tau \pm \left(\frac{\phi_\xi}{x_\xi} - \frac{\phi_\gamma \zeta_\xi}{x_\xi \zeta_\gamma}\right)\sqrt{(1+\zeta)} \ = \ 0 \tag{A11}$$

$$\zeta_\tau \pm \sqrt{(1+\zeta)}(\zeta_\xi/x_\xi) \ = \ 0 \tag{A12}$$

If the solid boundary grid is orthogonal (such as the seabed, bottom and wall of the cylinder, and side boundary in the $y$-direction), the boundary solution can be directly obtained from the Neumann condition of Equation (5).

*Appendix A.3. Numerical Discretization*

The transformed equations are discretized by the finite difference method. The details of the numerical discretization process can be found in work by Chang and Wang [68]. The vital processes are briefly described here; Equation (1) solves the points ($i$, $j$, $k$) in an element (Figure A1). Using the central difference method, the point solution can be arranged as:

$$\phi_{i,j,k} = \left\{ \begin{array}{l} C_{22B}\phi_{i,j,k-1} \quad +C_{22T}\phi_{i,j,k+1} + C_{32C}\phi_{i+1,j,k} + C_{12C}\phi_{i-1,j,k} + C_{23C}\phi_{i,j+1,k} + C_{21C}\phi_{i,j-1,k} \\ \quad +C_{32T}\phi_{i+1,j,k+1} + C_{12T}\phi_{i-1,j,k+1} + C_{32B}\phi_{i+1,j,k-1} + C_{12B}\phi_{i-1,j,k-1} + C_{23T}\phi_{i,j+1,k+1} \\ \quad +C_{21T}\phi_{i,j-1,k+1} + C_{23B}\phi_{i,j+1,k-1} + C_{21B}\phi_{i,j-1,k-1} \end{array} \right\} \tag{A13}$$
$$/\{2(g^{11} + g^{22} + g^{33})\}$$

In which,

$$C_{32C} \ = \ C_{12C} \ = \ g^{11}; \ C_{22T} \ = \ g^{33} + \frac{f^3}{2}; \ C_{22B} \ = \ g^{33} - \frac{f^3}{2};$$

$$C_{23C} \ = \ C_{21C} \ = \ g^{22}; \ C_{23T} \ = \ C_{21B} \ = \ \frac{g^{23}}{2}; \ C_{21T} \ = \ C_{23B} \ = \ \frac{-g^{23}}{2};$$

$$C_{32T} \ = \ C_{12B} \ = \ \frac{g^{13}}{2}; \ C_{12T} \ = \ C_{32B} \ = \ \frac{-g^{13}}{2}.$$

The subscription symbols *T*, *C*, and *B* in Equation (A13) indicate the grid coefficients at the upper, middle, and lower levels, respectively (as labeled in Figure A1).

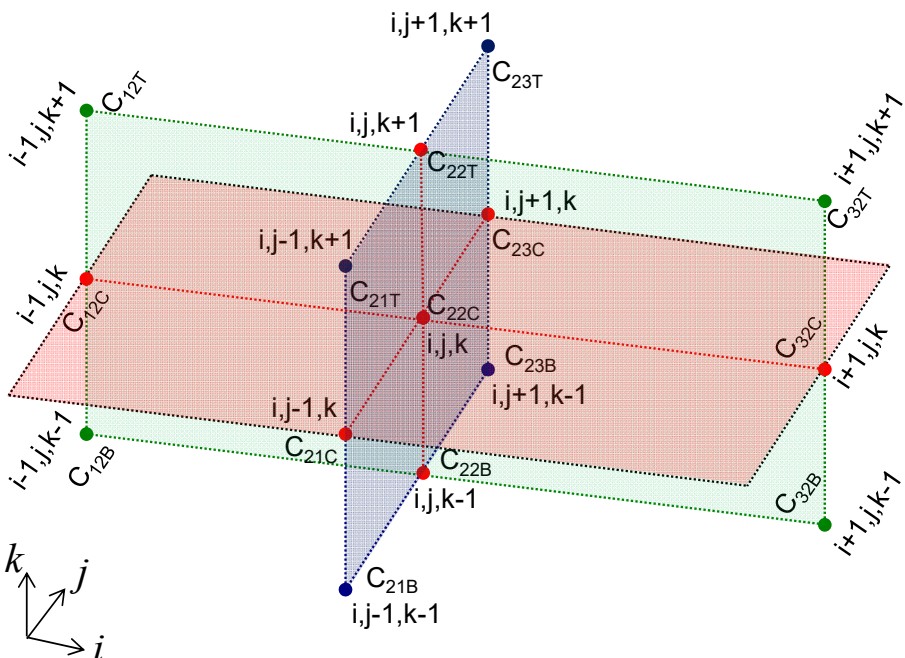

**Figure A1.** Schematic diagram of the numbered adjacent nodes in a grid element.

For free-surface boundary conditions, the first-order backward in the $\gamma$ derivative and first-order central difference in the $\xi$ and $\eta$ derivatives are adopted. For $x$-lateral boundary conditions, the first forward and first backward methods are applied to the left and right truncated sections, respectively, in the computational domain. The finite difference in the time derivative can be solved explicitly and implicitly, and then the average value is used as the correct result. Based on these principles, the differential formula for the dummy variable $Q$ is expressed by the operator as:

$$\delta_\xi Q = \frac{Q_{i+1,j,k}^{n+1} - Q_{i-1,j,k}^{n+1} + Q_{i+1,j,k}^{n} - Q_{i-1,j,k}^{n}}{4\Delta\xi} \tag{A14}$$

$$\delta_\eta Q = \frac{Q_{i,j+1,k}^{n+1} - Q_{i,j-1,k}^{n+1} + Q_{i,j+1,k}^{n} - Q_{i,j-1,k}^{n}}{4\Delta\eta} \tag{A15}$$

$$\delta_\gamma Q = \frac{Q_{i,j,k}^{n+1} - Q_{i,j,k-1}^{n+1} + Q_{i,j,k}^{n} - Q_{i,j,k-1}^{n}}{2\Delta\eta} \tag{A16}$$

$$\delta_\tau Q = \frac{Q_{i,j}^{n+1} - Q_{i,j}^{n}}{2\Delta\tau} \tag{A17}$$

$$\delta Q = \frac{Q_{i,j}^{n+1} + Q_{i,j}^{n}}{2} \tag{A18}$$

The superscript $n$ in the formula represents the time step, $\Delta\xi = \Delta\eta = 1$, and $\Delta\tau$ is the calculation time interval. The free-surface boundary conditions are discretized using an explicit-implicit hybrid method. The numerical calculation process uses the free-surface kinematic boundary condition to solve $\zeta$ and the free-surface dynamic boundary condition to solve the boundary value $\phi$. That is, Equations (A6) and (A7) can be expressed as:

$$\zeta_{i,j}^{n+1} = \zeta_{i,j}^{n} + \Delta\tau\left\{\delta w - \delta u\left(\frac{\delta_\xi \zeta}{\delta_\xi x}\right) - \delta v\left(\frac{\delta_\eta \zeta}{\delta_\eta y}\right)\right\} \tag{A19}$$

$$\phi_{i,j}^{n+1} = \phi_{i,j}^n + \Delta\tau\left\{\delta w\delta_\tau\zeta - \frac{1}{2}\delta\left(u^2 + v^2 + w^2\right) - \delta\zeta\right\} \tag{A20}$$

The opening boundary values can be derived from Equations (A11) and (A12) and then expressed as:

$$\begin{cases} \phi_{IM,j,k}^{n+1} = \phi_{IM,j,k}^n + \Delta\tau\left\{\delta w\delta_\tau\zeta - \sqrt{1 + \zeta_{IM,j}^{n+1}}\left(\frac{\delta_\xi\phi}{\delta_\xi x} - \frac{\delta_\gamma\phi\delta_\xi\zeta}{\delta_\xi x\delta_\gamma\zeta}\right)\right\} \\ \phi_{1,j,k}^{n+1} = \phi_{1,j,k}^n + \Delta\tau\left\{\delta w\delta_\tau\zeta + \sqrt{1 + \zeta_{1,j}^{n+1}}\left(\frac{\delta_\xi\phi}{\delta_\xi x} - \frac{\delta_\gamma\phi\delta_\xi\zeta}{\delta_\xi x\delta_\gamma\zeta}\right)\right\} \end{cases} \tag{A21}$$

$$\begin{cases} \zeta_{IM,j}^{n+1} = \zeta_{IM,j}^n - \Delta\tau\left\{\sqrt{1 + \zeta_{IM,j}^{n+1}}\left(\frac{\delta_\xi\zeta}{\delta_\xi x}\right)\right\} \\ \zeta_{1,j}^{n+1} = \zeta_{1,j}^n + \Delta\tau\left\{\sqrt{1 + \zeta_{1,j}^{n+1}}\left(\frac{\delta_\xi\zeta}{\delta_\xi x}\right)\right\} \end{cases} \tag{A22}$$

An over-relaxation iteration factor (SOR = 1.2~1.5) complements the solution process to accelerate iteration convergence. In this paper, the grid spacing $\Delta \approx 0.05 \sim 0.25$ is adopted, and the calculation time interval is $\Delta\tau = 0.05\sim0.1$. The numerical calculation process requires the convergence criteria $\left|\Omega^k - \Omega^{k-1}\right| \le 10^{-6}$ at each time step, where $\Omega$ can be $\phi$ or $\zeta$, and here the superscript $k$ means the $k$th iteration of the loop.

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
