# Peer review of "Interaction of a Solitary Wave with Vertical Fully/Partially Submerged Circular Cylinders with/without a Hollow Zone"

_jmse, doi:10.3390/jmse8121022_

Round 1
Reviewer 1 Report
Author took into account my remarks.
Reviewer 2 Report
The author has addressed most of my comments and the paper can be now published. However, the figures must be improved. Some are too big, some are spread over two pages. Consistency would be very much appreciated.This manuscript is a resubmission of an earlier submission. The following is a list of the peer review reports and author responses from that submission.
Round 1
Reviewer 1 Report
See attached file

Reviewer 2 Report
This paper presents a 3D fully nonlinear potential wave model to calculate the interaction of a solitary wave with a vertical cylinder piercing the water at different depths with or without hollow zones.
The paper is of interests for the readers of the journal. However, it has the following major flaws:
- The scientific novelty of the study is unclear and not specified at all.
- The scientific knowledge gap that the study aims to fill is not discussed
- The engineering problem that the study attempt to solve is not clear at all.
- The manuscript needs an overall proofreads for the use of the English. There are many sentences in which the use of pronouns make difficult to understand which is the subject.
- In the Introduction, a few times I read “this article…” but it is not clear at all if the Author
- The equations provided in section 2 are introduced without explaining if they are taken from a textbook or existing bibliography or if the equations are part of the innovative contribution from the Author.
On minor notes:
- Each figure should be in a single page rather than scattered across two pages;
- The font size used in Fig. 7 is too small in comparison to the font size used in other figures. In the same figures, the units are missing;
- The conclusions should not be in a bullet form and the path for future research should be considered as a single paragraph at the very end.
- A quick look at Scopus, Google Scholar, etc… revealed the publication of hundreds of papers on similar subjects. Under this light, the citation of 22 references only seems an incomplete account of the state-of-the-art knowledge.
In summary, I acknowledge and respect the work conducted by the Author I cannot recommend the publication of this manuscript in this a peer-reviewed journal.